# Metagenomic Analyses Reveal the Influence of Depth Layers on Marine Biodiversity on Tropical and Subtropical Regions

**DOI:** 10.3390/microorganisms11071668

**Published:** 2023-06-27

**Authors:** Bianca C. F. Santiago, Iara D. de Souza, João Vitor F. Cavalcante, Diego A. A. Morais, Mikaelly B. da Silva, Matheus Augusto de B. Pasquali, Rodrigo J. S. Dalmolin

**Affiliations:** 1Bioinformatics Multidisciplinary Environment—IMD, Federal University of Rio Grande do Norte, Natal 59078-400, Brazil; bianca.santiago.110@ufrn.edu.br (B.C.F.S.); iara.souza.065@ufrn.edu.br (I.D.d.S.); joao.cavalcante.073@ufrn.edu.br (J.V.F.C.); arthur.vinx@gmail.com (D.A.A.M.); 2Food Engineering Department, Federal University of Campina Grande, Campina Grande 58401-490, Brazil; mikaellyb66@gmail.com (M.B.d.S.); matheus.augusto@professor.ufcg.edu.br (M.A.d.B.P.); 3Department of Biochemistry—CB, Federal University of Rio Grande do Norte, Natal 59078-970, Brazil

**Keywords:** metagenome analysis, marine biodiversity, depth layers, pelagic zones

## Abstract

The emergence of open ocean global-scale studies provided important information about the genomics of oceanic microbial communities. Metagenomic analyses shed light on the structure of marine habitats, unraveling the biodiversity of different water masses. Many biological and environmental factors can contribute to marine organism composition, such as depth. However, much remains unknown about microbial communities’ taxonomic and functional features in different water layer depths. Here, we performed a metagenomic analysis of 76 publicly available samples from the Tara Ocean Project, distributed in 8 collection stations located in tropical or subtropical regions, and sampled from three layers of depth (surface water layer—SRF, deep chlorophyll maximum layer—DCM, and mesopelagic zone—MES). The SRF and DCM depth layers are similar in abundance and diversity, while the MES layer presents greater diversity than the other layers. Diversity clustering analysis shows differences regarding the taxonomic content of samples. At the domain level, bacteria prevail in most samples, and the MES layer presents the highest proportion of archaea among all samples. Taken together, our results indicate that the depth layer influences microbial sample composition and diversity.

## 1. Introduction

Marine biodiversity consists of the variety of life present in the sea at different levels of complexity, from taxa to ecosystems [1]. Even though the oceans cover 71% of the earth’s surface [2], only 16% of the planet’s described species are marine [3]. It is believed that most of the undescribed species are found in the marine environment [4] and the exploration of deep-sea areas can increase new species discovery [5]. The scientific literature on deep-water oceanic biodiversity explores topics such as the links between ecosystems and productivity. Between 1993 and 2020, 1287 publications on deep water biodiversity were retrieved from the Scopus database, with microbial analysis being one of the established and emerging research topics [6]. Given this scenario, new research projects and investigations of marine life arise, resulting in new information and questions about the organisms of this system, from its taxonomic classification to the interaction of the microbial community. In the last decades, open sea expeditions provided relevant amounts of ocean data, especially microbiome data.

A successful example is the Tara Oceans expedition [7], which investigated 35,000 samples with millions of protein-coding nucleotide sequences in 210 collection stations in 20 biogeographic provinces and collected samples of seawater and plankton from the Red Sea, Mediterranean Sea, Atlantic Ocean, Indic Ocean, Pacific Ocean, and Southern Ocean [8,9]. The Tara Oceans expedition was composed of a multidisciplinary team of more than 100 specialist researchers, in which an exclusive sampling program was implemented, covering optical genomic methods in order to describe viruses, bacteria, archaea, protists, and metazoans in their physical-chemical environment. The project stands out in the investigation of global oceanic biodiversity by integrating genetic, morphological, and functional diversity in its environmental context on a global oceanic scale and at multiple depths [10]. Data from the Tara Oceans expedition has been extensively studied in the past, having provided novel insights on marine viral communities [11,12], as well as showed the environmental drivers of microbial composition in distinct oceanic regions and depths [7].

Marine microorganism composition varies depending on latitude and depth [13]. Although earlier studies have described the vertical variation in microorganism diversity and abundance [14,15], most studies focus on bacteria and no study has provided a comprehensive exploratory analysis of the influence of depth on microorganism composition in tropical and subtropical regions. The “snapshot” sampling strategy, consisting of sampling at a single point in time at sparse sampling stations, is insufficient when considering the global ocean [10]. However, according to the authors, the Tara Oceans expedition uses strategies such as near real-time remote sensing—in addition to other data—to locate oceanographic features and strengthen ecosystem comparisons [8]. Tara Oceans project collected samples from three depth layers: the surface water layer (SRF-between 3 m and 7 m deep), deep chlorophyll maximum layer (DCM-between 7 m and 200 m deep), and mesopelagic zone (MES-between 200 m and 1000 m deep). The sea microbiome is complex, composed of viruses, bacteria, archaea, and unicellular eukaryotes, which are responsible for the impulse for the biogeochemistry global cycles that regulate the Earth system. While the small viruses are ubiquitous and the plankton more abundant in seawater, prokaryotes do 30% of primary production and 95% of community respiration in oceans [8].

With advances in next sequencing technologies, microbial ecology has advanced in classical environmental studies [16]. By September 2020, 429,731,711 DNA and RNA sequences, 2,401,136 taxonomic classifications and nomenclature catalogs, and 55,580 genome sequencing projects per organism had already been registered [17]. Metagenomic analysis emerged as a solution to deeply investigate complex microbial communities, such as those from marine environments [18]. Metagenomics allowed the researchers to study the genomes of most microorganisms that cannot be easily obtained in pure culture [19]. The use of genome-based approaches results in information about the metabolic potential and several phylogenetically informative sequences, which in turn can be used for the classification of organisms [20]. With metagenomics, it became possible to study microbial communities present in complex ecosystems regardless of cultivation, such as soil, ocean water, and the human body [21], and to evaluate the taxonomic and functional profiles of microbiomes [22]. One of the main approaches is shotgun sequencing, capable of performing microbial DNA sequencing without selecting a particular gene [23].

Here, we investigated microbiological communities from eight different collection stations in the tropical and subtropical zones of the globe, in three different depth layers. From the publicly available metagenomics data provided by Tara Oceans, we selected collection stations present in the tropical and subtropical belts of the globe that presented samples in the three depth layers simultaneously. The samples were grouped, taxonomically classified, and functionally annotated with the MEDUSA pipeline [24]. Clustering analysis of sample diversity separates the MES layer from the others. We also found that SRF and DCM depth layers show considerable similarity in terms of composition, abundance, and diversity, while the MES layer presents significantly greater diversity than the others.

## 2. Materials and Methods

### 2.1. Data Selection

The Tara Oceans project investigated the ocean’s plankton communities by using new and traditional methods to collect and analyze sea samples. The marine plankton collected represents organisms of six orders of magnitude in size, such as viruses, giruses, prokaryotes, protists, and zooplankton. To account for the different microorganisms’ sizes, seawater was filtered using different strategies, which effectively identified microorganisms in the following size fractions: <5 μm (or <3 μm), 5–20 μm (or 3–20 μm), <20 μm, 20–180 μm and 180–2000 μm for planktonic viruses, prokaryotes and unicellular eukaryotes, and >50 μm, >200 μm, >300 μm, >500 μm and >680 μm for large planktonic unicellular eukaryotes and metazoans [8]. The sampling was performed in collection stations that spanned different oceans and latitudes. Also, the samples were acquired from three depth layers: (i) surface water layer (SRF, between 3 m and 7 m deep), (ii) deep chlorophyll maximum layer (DCM, between 7 m and 200 m deep), and (iii) mesopelagic zone (MES, between 200 m and 1000 m deep). Sample collection lasted 24–48 h in each station and researchers took special care to reposition the SV Tara long schooner to avoid disturbances and ensure a homogenous ecosystem during collection, although environmental and sea conditions could have impacted the sampling process [8]. The SRF sampling occurred during the daytime on the first day and used the High Volume Peristaltic pump (HVP-pump) or the 10 L buckets; the DCM sampling was performed during the daytime on the second day, using the HVP pump or the Rosette Vertical Sampling System (RVSS); and the MES sampling was performed during the daytime on the first day using multiple RVSS deployments [8]. After multiple steps of seawater collection by HVP-pump and RVSS methods, the final volume taken was 3 L.

After collection, samples were processed by a variety of onboard and on-land methods, including flow cytometers and confocal microscopy, FlowCams, ZooScans, and electron microscopy. In addition, high-throughput sequencing methods were used to explore the phylogenetic content and the functional profiles of samples [8,10]. Following the different organism sizes, different DNA extraction protocols were used [12,25,26]. Then, 30 to 50 ng of DNA was sonicated, fragments were end-repaired, 3’-adenilated, adapters were added, and fragments were PCR-amplified as described in [7]. Libraries were sequenced in Illumina sequencers [7].

Here, we downloaded the publicly available shotgun metagenomic data from the Tara Oceans project at the European Nucleotide Archive (ENA), a repository of free and unrestricted access to annotated DNA and RNA sequences (https://www.ebi.ac.uk/ena/browser/downloading-data, accessed on 7 June 2023). We selected a set of tables available on the Tara Oceans archives website, in the tab Companion Website Tables (http://ocean-microbiome.embl.de/companion.html, accessed on 7 June 2023). We selected collection stations in the tropical and subtropical regions that contained at least one sample for each depth layer. Seventy-six samples were selected, collected between July 2010 and June 2011, from 8 tropical and subtropical collection stations. The selected stations are spread across the Indian Ocean (stations 064 and 065), the South Atlantic Ocean (stations 068, 076, and 078), and the South Pacific Ocean (stations 098, 111, and 112), encompassing a broad distribution of the global ocean microbiome. Each selected station presents data from the SRF, DCM, and MES layers. Appendix A shows the number of samples in each station by depth.

### 2.2. Metagenomics Analysis Workflow

We used MEDUSA, an open-source bioinformatics pipeline that gathers state-of-the-art tools for preprocessing and analysis of shotgun metagenomics data (https://github.com/dalmolingroup/MEDUSA, accessed on 7 June 2023) [24]. The MEDUSA pipeline has been reported to be both accurate and sensitive to taxonomic classifications and provides a functional annotation to metagenomics data [24]. This pipeline consists of four steps: pre-processing, alignment, taxonomic classification, and functional annotation.

#### 2.2.1. Pre-Processing

Raw data in FASTQ format were downloaded from the ENA database. The fastp tool (version 0.20.1) was used to remove adapters and low-quality sequences [27], as well as to merge paired-end sequences into single-end ones, a necessary step for the subsequent alignment step with Diamond. Fastp was executed with automatic adapter detection enabled and a minimum qualifying Phred score of 20 (–detect_adapter_for_pe -q 20), and merged reads were generated using the –merged_out parameter. Then, the FastX-Collapser tool (version 0.0.14) was used to collapse fully duplicated sequences present in the single-end samples resulting from the previous step [28], in order to reduce the size of the input data. FastX-Collapser was executed with default parameters.

#### 2.2.2. Alignment

We downloaded the non-redundant (NR) protein bank of the National Center for Biotechnology Information (NCBI) (Accessed on 21 March 2021). DIAMOND (version 2.0.9) was used for the alignment step, providing an efficient and fast read alignment against a protein database [29]. The DIAMOND aligner creates an index for the NR protein database (‘diamond makedb’) and then samples are aligned against this database, in BLASTx mode. In the DIAMOND alignment, the ‘–top’ option was used in order to only report matches within the top 3% scores (‘diamond blastx –top 3’).

#### 2.2.3. Taxonomic Classification

Kaiju software (version 1.7.4) was used to perform the taxonomic classification of samples [30]. First, a local index was created based on the microbial subset of the NR protein database containing all proteins belonging to archaea, Bacteria, and viruses, the same database we used for the alignment (Access date of 21 March 2021). This was done by the kaiju-makedb command, which downloads a source database and the taxonomy files from the NCBI FTP server, converts them into a protein database, and builds the Kaiju index. Next, a protein database alignment was performed, and reads were directly attributed to a species or a strain. The kaiju-addTaxonNames command was used to add taxonomic clade-level information (superkingdom, phylum, class, order, family, genus, and species) for each aligned read. In case of ambiguity, when an identical amino acid sequence matches two or more different species of the same genus, reads were annotated to higher-level nodes in the taxonomic tree [30]. In each file, only the taxonomically classified reads (i.e., the reads properly attributed to a known taxon) were considered for further analyses. The abundance is the number of reads attributed to a given taxon. Therefore, the higher the number of reads attributed to a given taxon in one sample, the higher the abundance of this sample.

#### 2.2.4. Functional Annotation

Since the alignment output provided hits to the NCBI NR database, we had RefSeq identifiers for each alignment hit. To better integrate these RefSeq identifiers with the Gene Ontology (GO) database, using the id mapping file provided by Uniprot (https://ftp.uniprot.org/pub/databases/uniprot/current_release/knowledgebase/idmapping/, accessed on 22 March 2021), we compiled a dictionary that mapped these RefSeq IDs to UniProt identifiers, which could then be mapped to GO identifiers. Using the annotate tool provided by MEDUSA [24], these dictionaries were used to map the best alignment hit of each sequence to the NCBI-NR database to its corresponding GO identifier.

### 2.3. Biological Diversity Analysis

We investigated the microbiome biodiversity by evaluating the diversity and abundance indices across different stations and depth layers in tropical/subtropical oceans. Diversity was estimated through the Shannon-Wiener Index (SWI) for each sample [31,32]. SWI is influenced by sample richness, i.e., the number of different species, as well as sample evenness, i.e., the species distribution uniformity [33]. As the richness and the evenness increase in a given sample, SWI also increases. SWI was calculated as the following. Consider $p_{ij}$ the proportion of species *i* in sample *j*. The total number of species in sample *j* is $S_{j}$. Therefore, the SWI for sample *j* is given by $H_{j}$ (Equation (Equation 1)):(1)$$H_{j}=-\sum_{i=1}^{S_{j}}p_{i}\log p_{i}$$

To compare SWI among the samples, SWI was normalized by the total number of species across all samples (*M*) (Equation (Equation 2)). Therefore, the diversity index for each sample is represented by the normalized SWI.
(2)$$H_{j}^{\prime }=-\frac{H_{j}}{\log M}$$

Additionally, the abundance index was estimated as the total number of reads for a given taxon in a given sample.

### 2.4. Clustering Analysis

We carried out an unsupervised clustering analysis to investigate microbial diversity across the samples from different stations and layers. To achieve this, we employed the unweighted pair group method with arithmetic mean (UPGMA), a hierarchical clustering algorithm that uses the pairwise distance matrix to cluster samples based on their normalized SWI values. This method calculates the Euclidean distance between any pair of samples, resulting in groups of samples hierarchically similar based on their diversity profiles. All 76 samples present in our original dataset were used for clustering analysis. The dissimilarity matrix produced can be found in the Appendix A.

The optimal number of clusters was defined by the Mojena method [34]. In this approach, the height of the dendrogram fusion points is used to determine the $\theta_{k}$ estimator, calculated by:$$\theta_{k}\;=\;\bar{\alpha }+k\hat{\sigma_{\alpha }}$$
where $\bar{\alpha }$ and $\hat{\sigma_{\alpha }}$ are the mean and the standard deviation for the height of the dendrogram fusion points, respectively; and k is a constant. The number of groups is determined when $\alpha_{j}>\theta_{k}$, considering $\alpha_{j}$ the values for fusion points distances, with j=1,...,n and *n* the sample size. We used k=1.25 to obtain the optimal number of clusters [35].

### 2.5. Functional Analysis

We primarily focused on exploring GO terms related to biological processes, and in particular, we aimed to identify the terms that were most distinct between different layers of our data. In order to visualize these differences, we calculated the frequency, that is, the total number of term occurrences, in each layer and represented the 50 most frequent terms in each layer with a heatmap, generated with the pheatmap R package (v1.0.12), using Euclidean distance and the “complete” method to hierarchically cluster the three layers (SRF, DCM, and MES) by the functional terms annotated.

## 3. Results

### 3.1. Biological Diversity Analysis

Here, we investigate the microbiological composition of three depth layers (SRF, DCM, and MES) in eight collection stations distributed among the Indian, the South Atlantic, and the South Pacific oceans (Appendix A). There is a noticeable heterogeneity between the samples (Figure 1). Figure 1A compares the diversity against the abundance by sample, where the abundance is the number of reads attributed to a given taxon, and the diversity is the normalized SWI (Equation (Equation 2)). Sample abundance ranged from 51,666 reads to 39,643,916 reads, with a median of 4,475,369 reads (Appendix A). The microbiome diversity is measured by Shannon-Wiener Index (See Section 2 (Materials and Methods) for details). The diversity index ranged from 0.40 to 0.64, with samples from the MES layer showing greater diversity than the samples from SRF and DCM layers (Figure 1A, Appendix A). As the majority of stations provided multiple samples, we averaged the diversity and abundance indices by each station. The MES layer stands out showing greater diversity than the other layers (Figure 1B, Appendix A). Additionally, the MES layer is consistently more diverse when considering the diversity index across stations (Figure 1C, Appendix A). The abundance distribution is wide, with samples diverging in orders of magnitude from each other in the same station and in the same layer (Figure 1D, Appendix A). By layer, the MES diversity distribution differs from the SRF and DCM layers (Wilcoxon rank sum test, adjusted *p*-value < 0.001, Figure 2A), however, there is no significant difference in abundance distribution between the layers (Figure 2B).

### 3.2. Clustering and Taxonomic Analysis

Samples were clustered by diversity. The unsupervised analysis identified seven groups of samples, named from G1 to G7 (Figure 3A). Groups were composed of samples from different oceans, except for G1 and G5, which were composed of unique samples. This indicates that the collection location may not be decisive to sample composition. When considering the depth layers, almost all MES samples were clustered in G6, while other groups are composed of both SRF and DCM samples (Figure 3A), reinforcing our previous observation of biological diversity distinction among MES and both SRF and DCM.

The taxonomic content of groups reflects the layer diversity differences. Figure 3B shows the proportion of domains (archaea, viruses, and bacteria) identified for samples in each group (Figure 3B). Bacteria prevail in groups G4 to G7. Samples from the G1 to G3 groups and a few samples from the G4 group show a high proportion of viruses. Samples from G6 show collectively the highest proportion of archaea among all groups. At the species level, the sets of overrepresented species differ between the groups (Figure 3C). Additionally, the proportions of each domain differ between the depth layers (Figure 3D). The proportion of archaea increases as the depth increases, with a higher proportion of archaea in DCM when compared to SRF (Wilcoxon rank sum test, adjusted *p*-value = 6.7 × 10${}^{-5}$) and in MES when compared to DCM (Wilcoxon rank sum test, adjusted *p*-value = 8.2 × 10${}^{-8}$). In contrast, deeper layers show a decreasing proportion of bacteria (SRF to DCM, Wilcoxon rank sum test, adjusted *p*-value = 3.6 × 10${}^{-3}$) and viruses (DCM to MES, Wilcoxon rank sum test, adjusted *p*-value = 2.6 × 10${}^{-8}$). Also, we observed significant proportion differences of viruses and bacteria between South Atlantic and Indian Oceans, and between South Pacific and South Atlantic Oceans (Appendix A).

### 3.3. Functional Content by Layer

We performed an exploratory analysis of layer functional composition. We evaluated the absolute frequency of GO terms annotated for each alignment sequence. Term frequency in SRF, DCM, and MES layers are represented in Figure 4 and in Appendix A. Terms such as *DNA replication*, *translation*, *transmembrane transport*, *DNA repair*, *tricarboxylic acid cycle*, *carbohydrate metabolic process*, and *glutamine metabolic process appear* in the three layers. These terms represent a core of essential biological functions common to the organisms in all layers. Additionally, we evaluated terms that are present exclusively in a given layer and that represent the layer-specific functional biological profile. Twenty-four, 53, and 10 terms appear exclusively in SRF, DCM, and MES layers, respectively. We also evaluated the distribution of exclusive terms identified in each station and by each layer (Appendix A). In spite of the greater diversity observed on the MES layer, there is no difference in the functional exclusivity between the layers. This indicates that the greater organism diversity observed in the MES layer might not be reflected in its functional diversity. The five most exclusive frequent terms in the SRF layer are *circadian rhythm*, *5-phosphoribose 1-diphosphate biosynthetic process*, *negative regulation of DNA recombination*, *metal ion transport*, and *sporulation resulting in formation of a cellular spore*. The five most exclusive frequent terms in the DCM layer are *heme-O biosynthetic process*, *intracellular protein transport*, *phosphate-containing compound metabolic process*, *cellular response to DNA damage stimulus*, and *vesicle-mediated transport*. Finally, the five most exclusive frequent terms in the MES layer are *methanogenesis*, *lysine biosynthetic process via aminoadipic acid*, *nitrate metabolic process*, *NADP biosynthetic process*, and *peptidyl-lysine modification to peptidyl-hypusine*.

## 4. Discussion

We evaluated the organismal and functional content of subtropical marine samples in three depth layers from eight collection stations from the Indian, the South Atlantic, and the South Pacific oceans. Abundance differences between samples from the same collection stations reflect high sample heterogeneity. Although no significant difference in abundance was observed between the three depth layers, we have observed diversity differences between each layer. It is hard to determine which factors (e.g., temperature, salinity, oxygen concentration, sunlight incidence, etc.) influence the diversity and to what extent. Nonetheless, similar patterns have been described in previous data showing that the influence of vertical changes in abundance, biomass, and species composition is more accentuated than the regional differences [36].

The mesopelagic (MES) layer is more diverse than the other layers, indicating that species found in deeper layers can adapt to low temperatures and lower incidences of sunlight. A principal component analysis of marine samples showed greater differences in community composition in MES compared to both SRF and DCM samples than the differences between SRF and DCM samples [7]. While the SRF layer was simply defined as a layer between 3 and 7 m below the sea surface and the DCM layer from the chlorophyll fluorometer, the MES layer, comprised between 200 m and 1000 m depth, was defined based on vertical profiles of temperature, salinity, fluorescence, nutrients, oxygen, and particulate matter [8]. This can lead to different conditions in this layer from season to season and, therefore, the sample selection criteria and the chemical composition of the mesopelagic zone can have an influence on the greater diversity identified at this depth.

Although the deep sea occupies 60% of the planet, the exploration of this region occurs in a much more restricted way than in coastal areas [3]. The deep sea provides an ideal scenario for species richness, given the less extreme temperatures than the ones observed in superficial layers [37]. A collection of heterogeneous habitats arise given the environmental conditions at bathyal depths (between 1000 m and 4000 m depth) and abyssal depths (from 4000 m depth) at different geographic and bathymetric scales [38,39]. However, the level of environmental heterogeneity has unknown effects on the dispersion and distribution of species both at high depths and in inland waters [40]. Thus, the species may have greater distribution ranges in deep waters than in shallow waters [3].

The deep sea—below 200 m deep—supports enormous biodiversity, thus being the largest biome on Earth, with 84% of the oceanic area below 2000 m deep [41]. Ocean environmental conditions are directly influenced by depth and topography, from light penetration to temperature and oxygen gradients [42]. Changes in ocean chemistry resulting from climate change are affecting even the most remote places in the ocean, and it remains to be seen whether the deep sea is resilient to anthropogenic disturbances [43]. With the advancement of autonomous technologies, the exploration of deep-sea areas can generate an increase in the discovery of species in this region [5].

Different patterns of marine chemical composition were observed at the different depths explored, in addition to a change in microbial community composition over time and between the different depths [44]. Clustering-based analysis indicates that metagenomic samples found in the third deep layer (MES) have similar and highly diverse microbial profiles. Analogously, the samples found in SRF and DCM have a similar average of diversity among themselves and between the layers, presenting significantly lower diversity indexes than in the MES layer. Bacteria are predominant in the majority of samples. Many factors contribute to the prokaryote diversity in ocean habitats, such as the availability of organic components and particulate matter, pelagic zones, and depth [15,45,46]. The variability of the number of bacteria is an issue addressed in several studies. In [47] it was observed that bacterial variability and abundance decrease with increasing water (1250–5600 m) and sediment depth (0–5 cm) in nine stations from the Fram Strait (Arctic Ocean). However, the most consistent explanation for this variability is that, regardless of depth, the most influential factor is the amount of organic material that reaches the bottom of the sea, together with its physical-chemical availability in situ [48]. In situ studies of deep-sea bacterioplankton showed the occurrence of bacterioplankton control by seafloor virioplankton, as there was a decrease in bacterial diversity and a change in its structure in the presence of diluted viruses [49]. Additionally, archaea communities were richer in deeper waters in the Arctic and Antarctic oceans, indicating that the more superficial archaea communities showed lower diversity than those in the deeper layers [50,51]. Despite the expected differences among tropical and subtropical regions, and Arctic and Antarctic regions, Jing and collaborators found that vertical variation patterns of diversity are similar in distinct latitudes of the Pacific Ocean, which included tropical and subtropical regions [14]. This could point to similar underlying mechanisms of the influence of depth in microorganism composition and abundance.

The concentration of viruses was high in 14 of the 76 samples studied here, surpassing Bacteria in five of these, all within samples from the most superficial layers of the study, SRF, and DCM. The viral abundance showed a significant interoceanic difference for the epipelagic (0 to 200 m depth) and mesopelagic (200 to 1000 m depth) zones [52]. Additionally, viral abundance has been shown to significantly decrease with depth in tropical and subtropical regions [52], which is consistent with our findings regarding groups from G1 to G3. In a metagenomic analysis of DNA virus diversity, it was observed that the most abundant and widespread viral Operational Taxonomic Units (vOTUS) in South China Sea DNA virome (SCSV) are from uncultured viruses annotated from viral metagenomics, indicating that most marine viruses have not yet been characterized [53]. Despite the abundance of Viruses pointed out by several studies, due to issues such as the fragility of the environmental viral gene bank, the genetic and biological diversity of aquatic viruses remains largely unexplored [54,55,56].

Our analysis revealed higher archaea proportions in the MES layer, which suggest that this clade may have an important contribution to the diversity of deeper layers. Despite this result, this subject is still under constant discussion, since the identification of the different components in microbial communities is only a preliminary step, still requiring data about their functions, interactions, spatial and temporal dynamics in addition to environmental parameters [57].

In our functional analysis, we quantified GO terms based on the alignment’s hits. By using the alignment results to perform functional annotation we have an intrinsic link between functional and taxonomic results since they utilize the same database. Although greater biological diversity was observed in the taxonomic result in MES layers, we could not ascertain a significant functional difference between the depth layers. We acknowledge that other methods could be advantageous, such as genome mining approaches [58], like the detection of biosynthetic gene clusters [13] and antimicrobial peptides [59].

Our results show that the three depth layers have similar relative abundances, irrespective of the geographic region (i.e., collection stations). Additionally, our analysis reveals that the mesopelagic layer exhibits the highest level of biodiversity in comparison to the other two layers, and that depth can have a significant impact on microorganism diversity. Moreover, although the organisms in all three depth layers share a set of common core functional profiles, our investigation highlights certain biological functions that are exclusive to each depth layer. This suggests that these layers possess unique characteristics and functional diversity which could be tied to environmental factors related to depth, such as oxygen availability. Furthermore, the higher diversity and functional exclusivity in deeper layers could point toward more complex and as of yet undiscovered microbial communities.

## Figures and Tables

**Figure 1 microorganisms-11-01668-f001:**
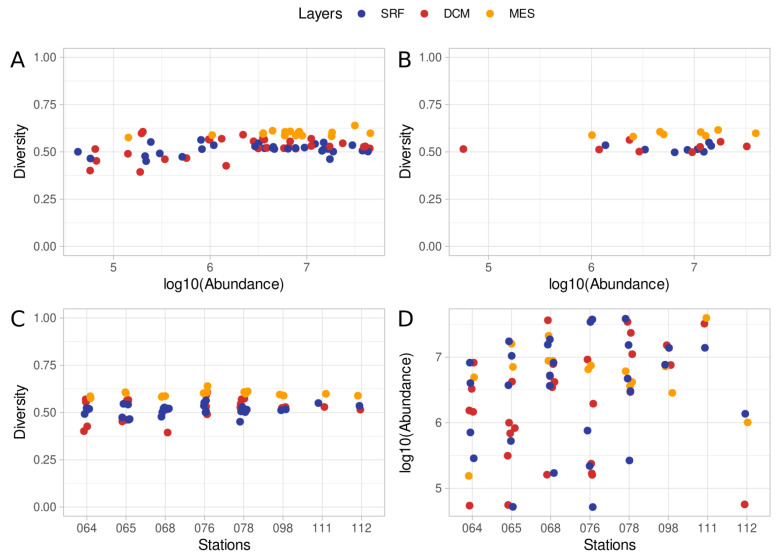
Distribution of diversity and abundance indices. (**A**) Comparison of diversity and log10-scaled abundance indexes by sample. (**B**) Diversity and log10-scaled abundance averages by the station. (**C**) Diversity distribution by the station. (**D**) Log10-scaled abundance distribution by the station. The abundance index is the number of reads attributed to a given taxon in a given sample. The diversity was estimated by the normalized Shannon-Wiener Index (SWI, see Section 2 (Materials and Methods) for details).

**Figure 2 microorganisms-11-01668-f002:**
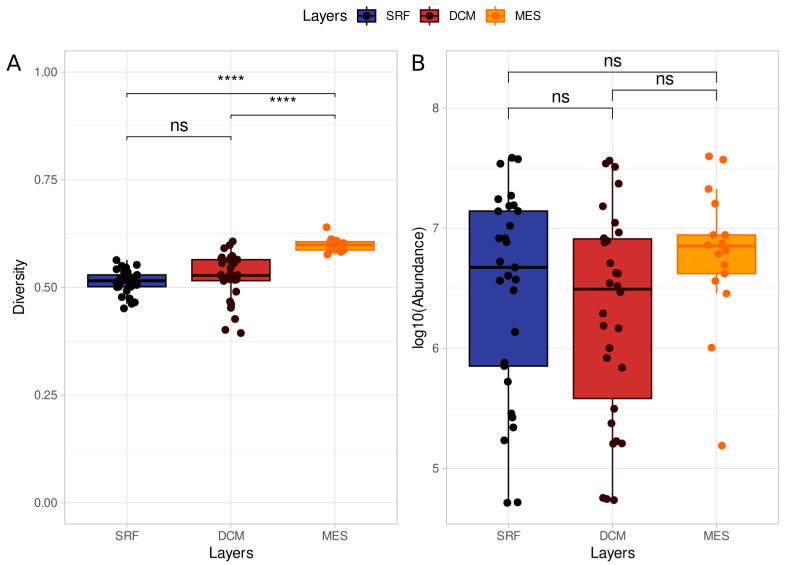
Diversity and abundance distribution by layer. (**A**) Diversity comparison between layers shows increasing diversity with increasing depth. (**B**) Abundance comparison between layers shows no significant association with depth layers. Comparisons were performed with the Wilcoxon rank sum test and Bonferroni-adjusted *p*-values were represented in each comparison. Significant comparisons are the ones with adjusted *p*-values < 0.05. Statistical significance is represented as follows. ns (adjusted *p*-value > 0.05); **** (adjusted *p*-value ≤ 0.0001). The abundance index is the number of reads attributed to a given taxon in a given sample. The diversity was estimated by the normalized Shannon-Wiener Index (SWI, see Section 2 (Materials and Methods) for details).

**Figure 3 microorganisms-11-01668-f003:**
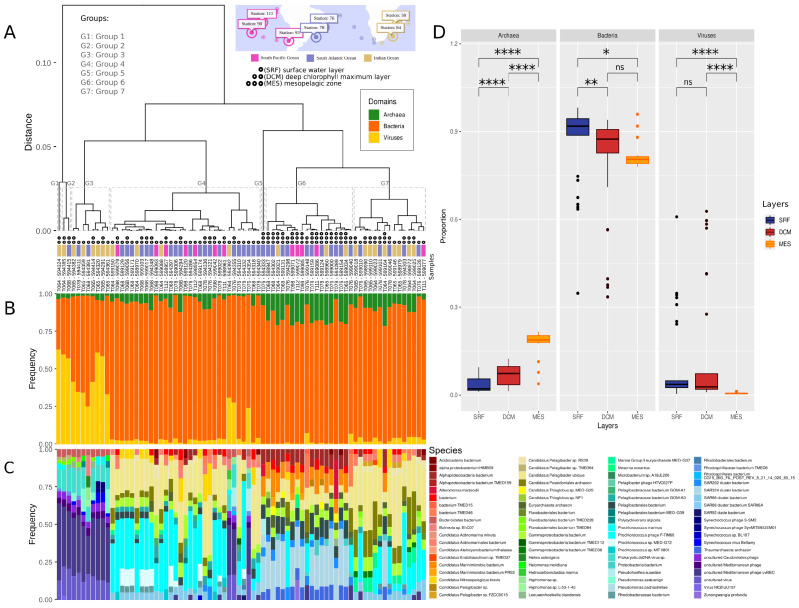
Clustering analysis of samples by diversity. (**A**) Dendrogram of samples clustered by the diversity index. Seven groups of samples were identified and named from G1 to G7. Samples are also represented by depth layers and oceans. The sample names are a combination of the station and the accession number. (e.g., the T064-594324 sample belongs to the station 064, and the ENA accession number is ERR594324). (**B**) The domain-level taxonomic classification shows the proportion of each domain (archaea, bacteria, and viruses) in each sample. (**C**) The species-level taxonomic classification shows the proportion of each species (the ten most frequent) in each sample. (**D**) Distribution of domain proportions considering the samples of each depth layer. Statistical significance is represented as follows. ns (adjusted *p*-value > 0.05); * (adjusted *p*-value ≤ 0.05); ** (adjusted *p*-value ≤ 0.01); **** (adjusted *p*-value ≤ 0.0001).

**Figure 4 microorganisms-11-01668-f004:**
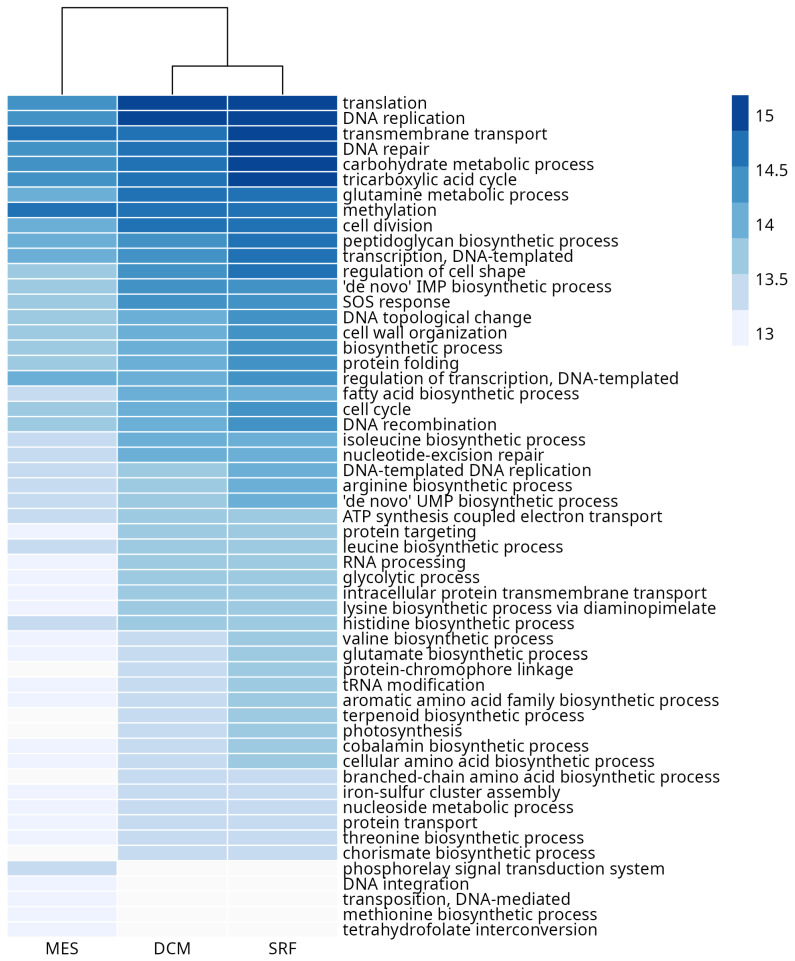
Frequency (log10-transformed) in each layer of the 50 overall most common functional terms. Frequency is defined as the total number of term occurrences in each layer (see Section 2 (Materials and Methods) for details).

## Data Availability

Scripts used for the analyses performed here can be found at: https://github.com/dalmolingroup/meta_taraoceans (accessed on 7 June 2023).

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
