# Peer review of "Metagenomic Analyses Reveal the Influence of Depth Layers on Marine Biodiversity on Tropical and Subtropical Regions"

_microorganisms, 2023, doi:10.3390/microorganisms11071668_

Round 1

Reviewer 1 Report

This study uses existing databases of metagenomic DNA sequence to perform analyses to determine factors that differentiate three strata ocean layers.  The analyses are exploratory but I think the authors could have a more detailed work, particularly in the area of pathway analysis which would have strengthened the paper considerably.

Major

 It is not clear how abundance was determined On line 175 the authors state "Additionally, the abundance index was estimated as the total number of organisms in a given sample."  but they do not state how it was measured.  In their reference 45 (Deming, J.W.; Carpenter, S.D, 2008), staining was used to enumerate bacteria for the determination of abundance.  Can the author provide additional details (a reference would be nice...)?

I believe the results of their analyses (that are depicted graphically in the paper ) should be included as Tables in supplementary information.

Minor

line 43    "according to the authors"  need a citation after authors...

line 55   change "limited to research on" to "of limited use for"

line 56   change "electable" to "able"

line 92-94   change "m" to "um"

line 312   change "Viruses had a high incidence" to "The concentration of viruses was high ..."

Figure 4.  It would be more useful to have this represented as a heatmap graph/table that allows the three regions tested to facilitate comparisons.  Even if this was done, however, it would be difficult to deduce what the differences mean.

I

A few minor changes are needed to improve clarity.

Author Response

REVIEWER 1

This study uses existing databases of metagenomic DNA sequence to perform analyses to determine factors that differentiate three strata ocean layers.  The analyses are exploratory but I think the authors could have a more detailed work, particularly in the area of pathway analysis which would have strengthened the paper considerably.

Authors' comment:

We thank the Reviewer for the revision he/she provided. We have implemented all of his/her suggestions, which substantially improved the paper. We have replaced Figure 4 following the Reviewer suggestion, which enriched the representation of the results regarding pathway analysis. 

Reviewer comment:

It is not clear how abundance was determined On line 175 the authors state "Additionally, the abundance index was estimated as the total number of organisms in a given sample."  but they do not state how it was measured. In their reference 45 (Deming, J.W.; Carpenter, S.D, 2008), staining was used to enumerate bacteria for the determination of abundance.  Can the author provide additional details (a reference would be nice...)?

Author’s response:

We agree that the abundance index was not clear enough in the original manuscript version and we thank the Reviewer for his/her comment. In our work, the abundance is the number of reads attributed to a given taxon in a sample. In the taxonomic classification step of MEDUSA, executed with the Kaiju software, reads are attributed to a species or a strain, or to a higher-level taxon in case of ambiguity, as described in the “Taxonomic classification” section of the Materials and Methods. Therefore, In Figure 1, for example, the abundance index for a sample is the total number of reads that were attributed to a given taxon. We extensively improved the manuscript (methods section, results section, and figure legends) to make it as clear as possible.

Reviewer comment:

I believe the results of their analyses (that are depicted graphically in the paper) should be included as Tables in supplementary information.

Authors’ response:

We agree with the Reviewer and we have provided more detailed supplementary information in spreadsheet format (Supplementary tables S2 to S5), including the data used to produce the figures. 

Reviewer comment: 

line 43    "according to the authors"  need a citation after authors…

line 55   change "limited to research on" to "of limited use for"

line 56   change "electable" to "able"

line 92-94   change "m" to "um"

line 312   change "Viruses had a high incidence" to "The concentration of viruses was high ..."

Authors’ response:

We thank the Reviewer for the suggestion and have made the alterations in the manuscript. 

Reviewer comment:

Figure 4. It would be more useful to have this represented as a heatmap graph/table that allows the three regions tested to facilitate comparisons. Even if this was done, however, it would be difficult to deduce what the differences mean.

Authors’ response:

We thank the Reviewer for the great suggestion and acknowledge that such visualization is challenging. We produced a heatmap of the 50 most frequent terms, which allows the direct comparison of terms identified for the three layers. In our opinion, the new figure substantially improves the result representation. 

Reviewer 2 Report

First of all it is not clearly explained whether the authors themselves did the metagenomic analysis or used a ready-made database. If they did it themselves, there is no detailed methodology described. If, on the other hand, they used the database, then why a description of the sampling methodology?   In Figures 1 and 2, the term diversity is used. It is not known what this indicator is. Is it S-H? Or maybe 1/S-H? Or maybe this indicator refers to something else?   In the text, the authors give values without units, e.g. line 198. This requires supplementation.   There is no clear conclusion at the end.  

Supplementary Table S1 - the table is divided into two pages, which makes it impossible to find important information. it needs improvement. Maybe it's worth changing the orientation of the table or reducing the font size?

In addition, data access dates should be added in Supplementary Table S1. (The same annotation applies to table 2 SM).

Author Response

REVIEWER 2

Reviewer comment:

First of all it is not clearly explained whether the authors themselves did the metagenomic analysis or used a ready-made database. 

Authors’ comment:

We thank the Reviewer for his/her comments. We addressed all the suggestions he/she made which improved the paper. In this manuscript, we analyzed publicly available metagenomic data provided by the Tara Oceans Project. As we described in the Materials and Methods (“data selection”) section, the list of samples can be accessed on the Tara Oceans website (http://ocean-microbiome.embl.de/companion.html) and the raw data can be downloaded on the ENA (https://www.ebi.ac.uk/ena/browser/downloading-data). We improved the Abstract and the Materials and Methods (“data selection”) sections to make them as clear as possible. 

Reviewer comment:

If they did it themselves, there is no detailed methodology described. If, on the other hand, they used the database, then why a description of the sampling methodology?

Authors’ comment: 

For transparency, we chose to summarize the sampling method used on the Tara Oceans project and depicted in the Pesant et al. paper (doi.org/10.1038/sdata.2015.23). This is relevant because sampling methodology is an important factor to be considered when analyzing the abundance, diversity, and functional differences between the ocean layers. For the full description of the sampling process, one can refer to the Tara Oceans methods paper (doi.org/10.1038/sdata.2015.23). 

Reviewer comment:

In Figures 1 and 2, the term diversity is used. It is not known what this indicator is. Is it S-H? Or maybe 1/S-H? Or maybe this indicator refers to something else?

Authors’s comment:

We thank the Reviewer for the observation. In this work, the diversity is estimated by the Shanon-Wiener index (SWI), which is also commonly described in the literature as S-H index. We described the SWI index in the “Biological diversity analysis” section in the Materials and Methods. In summary, for each sample, the proportion of species i in sample j is represented by pij and the total number of species in sample j is Sj. Then, the SWI for sample j is given by Hj:

To compare the SWI values across samples, the SWI was normalized by the total number of species across all samples (M), as the following:

To improve clarity, we added a sentence on Figure 1 and 2 legends specifying that the diversity is estimated by the normalized SWI.

Reviewer comment:

In the text, the authors give values without units, e.g. line 198. This requires supplementation.

Authors’ response:

We thank the Reviewer for the comment. The abundance index is the total number of reads attributed to a given taxon. We changed the main text accordingly to provide accurate information. We also added an explanation for the abundance and diversity definition on Figures 1 and 2, to facilitate the comprehension of the reader. 

Reviewer comment:

There is no clear conclusion at the end. 

Authors’ response:

The main contribution of our work is to show that the MES layers have different biodiversity properties when compared to the other layers in the evaluated samples. We also suggest by an exploratory analysis that these differences can be reflected in the layer's functional profile. We added a sentence in the final paragraph of the Discussion section to explicit the main implication of the results found in our work. 

Reviewer 3 Report

Manuscript is devoted metagenomic analyses of different layers of three oceans in tropical and subtropical regions. Authors used data obtained in Tara Ocean projects and accessible online for all. Nowadays a lot of similar articles are published. Metagenomic sequences and binning genomes allow us to study microbial community in different biotopes and their metabolism, understand how they interact with each other.

Usually researchers use the same set of methods, applications and pipelines. However, the authors of this study used non-standard data visualization methods (figure 4) and bioinformatics pipeline (MEDUSA), for which, unfortunately, there are only three publications in NCBI Pubmed. At the same time, there is only one for this application. Perhaps this is the main comment concerning this study. Perhaps the authors should have used additional application that would have supported the data obtained with using MEDUSA. Figure 4: interesting solution for data visualization, at first it is surprising. In my opinion, it is more applicable for social sciences.

Also I have comments. Since the authors used open data obtained by other scientists, it is unclear from the introduction whether the analysis of these data was carried out earlier. It is unclear for me: according to abstract, 8 stations and three horizons were taken (8x3=24). Why 76 samples? It is also unclear why these stations were chosen. It is also unclear: what volumes were taken for filtering, how DNA was extracted. At the same time, the authors reported some other details, for example, fractions of microorganisms. I think that either specify all point or give a link for all.

I think for readers it is better to see figure S2 with a diversity of microorganisms in the main body of manuscripts (sample names should be the same throughout the text).

Figure 1D – Why did the abundance vary so much in the one layer? Why were different numbers of samples taken for analysis of different layers?

Section 3.2. It is not clear how groups G1-G7 were obtained? Which stations were included in groups? Figure 3 is small, although it is just interesting in which ocean what group of microorganisms prevails?

Line 301-311 It is confusingly to use data about Arctic and Antarctic oceans for tropical and subtropical regions.

Author Response

REVIEWER 3

Manuscript is devoted metagenomic analyses of different layers of three oceans in tropical and subtropical regions. Authors used data obtained in Tara Ocean projects and accessible online for all. Nowadays a lot of similar articles are published. Metagenomic sequences and binning genomes allow us to study microbial community in different biotopes and their metabolism, understand how they interact with each other.

Authors’ comment: 

We thank the reviewer for the insightful suggestions he/she provided which substantially improved the paper.  

Reviewer comment:

Usually researchers use the same set of methods, applications and pipelines. However, the authors of this study used non-standard data visualization methods (figure 4) and bioinformatics pipeline (MEDUSA), for which, unfortunately, there are only three publications in NCBI Pubmed. At the same time, there is only one for this application. Perhaps this is the main comment concerning this study. Perhaps the authors should have used additional application that would have supported the data obtained with using MEDUSA. 

Authors’ response:

Starting with Figure 4, we immensely thank the reviewer for the comment and the suggestion. We produce a new Figure 4, which improves the data representation and interpretation. Regarding the MEDUSA pipeline, it was developed to gather the best tools designed for metagenomic analysis. MEDUSA orchestrates the gold-standard and widely used metagenomic tools, such as the DIAMOND, Kaiju, MEGAHIT, Samtools, BowTie2, fastp, and FasX-Collapser. Please see the table below to verify each tool and how cited they are. MEDUSA is built with Snakemake, a workflow management system highly popular for bioinformatics analysis. Also, MEDUSA is built in a modular way, meaning that all of the aforementioned tools can be used in a particular step of the analysis. For the ones with knowledge of workflow management and metagenomics analyses, the modules and their tools are completely tunable. Therefore, our results were obtained in accordance with the best practices used in the metagenomics analyses. 

Tool

Paper

Citations

fastp

https://doi.org/10.1093/bioinformatics/bty560 

7221

FastX-Collapser *

-

BowTie2

https://doi.org/10.1038/nmeth.1923 

39372

Samtools

https://doi.org/10.1093/bioinformatics/btp352 

45536

Kaiju

https://doi.org/10.1038/ncomms11257 

1268

DIAMOND

https://doi.org/10.1038/nmeth.3176 

7112

MEGAHIT +

https://doi.org/10.1093/bioinformatics/btv033 

3778

List of tools included as part of the MEDUSA pipeline, their papers, and their respective number of citations as of May 20th, 2023. Data gathered from Google Scholar.

* FastX-Collapser is not associated with any publication.

+ MEGAHIT, although not used in this paper, is included as the assembly step of the MEDUSA pipeline.

Reviewer comment:

Figure 4: interesting solution for data visualization, at first it is surprising. In my opinion, it is more applicable for social sciences.

Authors’ response:

Again, we agree with the Reviewer that such visualization is unusual for the life sciences. In order to improve the representation of the functional annotation, we have changed Figure 4 to be a heatmap showing term frequency in each layer.

Reviewer comment:

Since the authors used open data obtained by other scientists, it is unclear from the introduction whether the analysis of these data was carried out earlier. 

Authors’ response:

This data has been analyzed by other works, we acknowledge past analyses and have included a sentence in our Introduction detailing earlier findings using the Tara Oceans dataset.

Reviewer comment:

It is unclear for me: according to abstract, 8 stations and three horizons were taken (8x3=24). Why 76 samples? It is also unclear why these stations were chosen.

Authors’ response:

The aim of this work is to analyze the ocean microbiome data from subtropical/tropical regions. We chose the collection stations in these latitudes which contained at least one sample for each depth layer. In some stations, such as the 064, 065, and 068, each layer is represented by multiple samples. In others, such as 111 and 112, each layer is represented by only one sample. We have added a sentence in the “Data selection” section from the Materials and Methods to make it clear. 

Reviewer comment:

It is also unclear: what volumes were taken for filtering, how DNA was extracted. At the same time, the authors reported some other details, for example, fractions of microorganisms. I think that either specify all point or give a link for all.

Authors’ response:

We agree with the Reviewer and we have added the information and references for DNA extraction. Regarding the collection volumes, the main methodology paper for the Tara Oceans Project is not precise about the total amount of seawater collected initially. For each sampling strategy, different volumes of water were collected. For the of HVP-pump and RVSS collection methods, the final volume taken was 3 l. We added this description to the “Data selection” section of the Materials and Methods.  

Reviewer comment:

I think for readers it is better to see figure S2 with a diversity of microorganisms in the main body of manuscripts (sample names should be the same throughout the text).

Authors’ response:

We thank the Reviewer for the great suggestion. We moved the bar plot from the Supplementary Figure S2, which represents the ten most frequent species in each sample, to a new panel on the Figure 3 (now, panel C). Also, we added the sample names to the bars on the bar plot.

Reviewer comment:

Figure 1D – Why did the abundance vary so much in the one layer? Why were different numbers of samples taken for analysis of different layers?

Authors’ response:

In our analysis, the abundance is the total number of reads attributed to a given taxon. This index provides a proxy for the number of organisms in a given sample. However, the landscape of the organism composition in a given sample can be better evaluated when considering both the abundance and the diversity measures. This is because many technical and biological factors, such as the weather during the collection, the filtration process, and the library construction, can affect the number of reads being sequenced in each sample. We addressed the influence of technical factors at the end of the second paragraph of the Discussion section. As stated in the previous comment about the choice of the stations, we chose the collection stations in tropical and subtropical regions which contained at least one sample for each depth layer. Therefore, we considered all the samples from a given layer and a given station in order to improve the power of the statistical tests performed to compare the different layers. 

Reviewer comment:

Section 3.2. It is not clear how groups G1-G7 were obtained? Which stations were included in groups? Figure 3 is small, although it is just interesting in which ocean what group of microorganisms prevails?

Authors’ response:

We performed a hierarchical clustering on the diversity index calculated for each sample using the UPGMA method. For group selection, we applied the Mojena method, a simple and widely used algorithm to group definition on a dendrogram. We added a description of the algorithm to the “Clustering analysis” section of the Materials and Methods.

Reviewer comment:

Line 301-311 It is confusingly to use data about Arctic and Antarctic oceans for tropical and subtropical regions.

Authors’ response: 

We agree with the reviewer that comparisons between tropical and subtropical regions with arctic and antarctic ones can be misleading or confusing. However, since in this paragraph we are comparing the vertical variation patterns in abundance and diversity of microorganisms of our data to the findings of these earlier studies, we still think this comparison is valid. Jing et al, 2013 found that depth variations had a similar impact on diversity in three distinct latitudes of the Pacific Ocean, which included subarctic and tropical regions. Given these findings and the limited amount of studies on the influence of depth in tropical and subtropical regions, we chose to maintain the comparison. However, we have addressed the reviewer’s concerns by adding a sentence explaining the reason for this comparison.

Reviewer 4 Report

Review on “Metagenomic Analyses Reveal the Influence of Depth Layers on Marine Biodiversity on Tropical and Subtropical Regions” for Microorganisms (manuscript ID microorganisms-2399102)

The introduction describes some key aspects of marine metagenomics. Authors stress that genomic approached let to “deeply investigate complex microbial communities”. However, authors don’t justify the choice of the particular “8 different collection stations” (L76). The intro part L53–L74 is very common and could be omitted or shrunk. Please figure out the particular interest of the comparison between the microbiome diversity of various ocean layers in Intro and Discussion.

Unfortunately, the Supplementary files are missing, and I cannot to check the raw data or reproduce some of the results.

L31: maybe not “millions of microorganisms”, but millions of kinds/species?

My questions about Methods:

MEDUSA tool looks like a black box without tuning ability (according to the paper https://doi.org/10.3389/fgene.2022.814437 and repository https://github.com/arthurvinx/MEDUSA), which was tested only on synthetic datasets, not environmental ones. Tool released 2 years ago and no cites of the paper or “stars” of the repository despite the “MEDUSA outperformed MEGAN in these functional results”.

The distance criteria when dividing into the 7 groups (Fig. 3) is not described.

L131: Were the default settings used with fastp and FastX-Collapser? Please specify

L137: the date or version of the NR database does matter.

L160: how the dictionary was made? Is the script or dictionary itself are available? Figshare or Zenodo could help.

L178, L212: how the clustering was performed in details? The raw data (distance matrices better to be published as Supplementary table).

My questions about Results and Discussion:

How can authors interpret very high percentage of viruses (Fig. 3B)?

The signs of domains (Archae, Bacteria, Viruses) are hard to interpret and intersecting with the tree. Sample ID’s are missing on the Figure 3.

Colors of the same terms at different panels are different, which make them harder to compare.

L285: the reference [35] doesn’t contain the determination of “oceanic area below 2000 m deep”, please cite the original source.

L303: this manuscript describe the microbiome of “tropical or subtropical regions”, comparison with Arctic region looks strange here.

L316: virus taxonomy data are missing.

L319: the reference [51] describe the viral particles extraction protocol only, not the cause of “diversity of aquatic viruses remains largely unexplored”.

Some minor corrections to the text (style and spelling):

·        L271, 273: repetition, please rephrase

·        The references (24, 26, 47, 50) have missing journal name/DOI.

Author Response

REVIEWER 4

The introduction describes some key aspects of marine metagenomics. Authors stress that genomic approached let to “deeply investigate complex microbial communities”. However, authors don’t justify the choice of the particular “8 different collection stations” (L76). The intro part L53–L74 is very common and could be omitted or shrunk. Please figure out the particular interest of the comparison between the microbiome diversity of various ocean layers in Intro and Discussion.

Authors’ comment:

We thank the Reviewer for the revision he/she made. The aim of this work is to analyze the ocean microbiome data from subtropical/tropical regions. We chose the collection stations in these latitudes which contained at least one sample for each depth layer. In some stations, such as the 064, 065, and 068, each layer is represented by multiple samples. In others, such as 111 and 112, each layer is represented by only one sample. We have added a sentence in the “Data selection” section from the Materials and Methods to make it clear. We substantially shrunk the mentioned paragraph (i.e., L53–L74). We have also added a sentence in the third paragraph of the Introduction justifying why we explore the microorganism diversity and abundance between these different layers.

Reviewer comment:

Unfortunately, the Supplementary files are missing, and I cannot to check the raw data or reproduce some of the results.

Authors’ response:

We provided the raw data in abundance level for every taxon in every sample analyzed in our study (Supplementary Table S3). The summarized results used to produce Figures 1 and 2 (Supplementary Table S4), as well as the dissimilarity matrix from the clustering analysis (Supplementary Table S2), and the term frequency for each layer (Supplementary Table S5) were also added as Supplementary Tables in a spreadsheet format. 

Reviewer comment:

L31: maybe not “millions of microorganisms”, but millions of kinds/species?

Authors’ response:

We thank the Reviewer for noticing this. According to the paper of Sunagawa and collaborators (doi.org/10.1126/science.1261359), they were able to sequence millions of protein-coding nucleotide sequences. We changed the sentence in the L31 to provide the accurate information. 

Reviewer comment:

MEDUSA tool looks like a black box without tuning ability (according to the paper https://doi.org/10.3389/fgene.2022.814437 and repository https://github.com/arthurvinx/MEDUSA), which was tested only on synthetic datasets, not environmental ones. Tool released 2 years ago and no cites of the paper or “stars” of the repository despite the “MEDUSA outperformed MEGAN in these functional results”.

Authors’ response:

As pipelines that are managed by Snakemake, all the tasks performed during the pipeline’s run can be found in the rules described in a plain text file named Snakefile (https://github.com/arthurvinx/MEDUSA/blob/main/Snakefile). Each rule described in this file contains the exact commands executed via Shell Script during the run, including the arguments passed to the third-party tools, and an execution plan can be obtained with a Snakemake dry-run to check which rules will be performed according to the pattern found in the input files naming convention. Although the last change on the pipeline’s latest version was made two years ago, according to the repository commits, the paper was published on 07 March 2022, roughly 1 year and 2 months ago, and the tool was downloaded from the Anaconda Cloud 141 times (https://anaconda.org/arthurvinx/medusapipeline). The MEDUSA executes the metagenomics pipeline through the orchestration of gold-standard and widely used tools for metagenomics analysis, such as fastp, BowTie2, DIAMOND, Kaiju, Samtools, and MEGAHIT; all of them are highly cited among bioinformatics works (see Table below). Regarding the tuning, the MEDUSA original paper supplementary material describes the steps executed by the pipeline and arguments available for each tool, mentioning in the discussion that “To change the rules used during the pipeline execution, the user must edit the Snakefile”. Finally, besides the pipeline being tested with synthetic datasets for the paper, the DIAMOND aligner, a core tool on the pipeline’s workflow, is extensively used on environmental datasets by other authors published papers. Of course, as MEDUSA was developed as a Snakemake workflow, it is meant to be used by specialists. 

Tool

Paper

Citations

fastp

https://doi.org/10.1093/bioinformatics/bty560 

7221

FastX-Collapser *

-

BowTie2

https://doi.org/10.1038/nmeth.1923 

39372

Samtools

https://doi.org/10.1093/bioinformatics/btp352 

45536

Kaiju

https://doi.org/10.1038/ncomms11257 

1268

DIAMOND

https://doi.org/10.1038/nmeth.3176 

7112

MEGAHIT +

https://doi.org/10.1093/bioinformatics/btv033 

3778

List of tools included as part of the MEDUSA pipeline, their papers, and their respective number of citations as of May 20th, 2023. Data gathered from Google Scholar.

* FastX-Collapser is not associated with any publication.

+ MEGAHIT, although not used in this paper, is included as the assembly step of the MEDUSA pipeline.

Reviewer comment:

The distance criteria when dividing into the 7 groups (Fig. 3) is not described.

Authors’ response:

The distance criteria to cut the dendrogram and obtain the groups was done by the Mojena method, as mentioned in the “Clustering analysis” section of Materials and Methods. To improve clarity,  we added a description of the Mojena method in this section. 

Reviewer comment:

L131: Were the default settings used with fastp and FastX-Collapser? Please specify

Authors’ response: 

The parameters have now been detailed in the ‘Pre-processing’ section of our methodology. FastX-Collapser was executed with default parameters, while fastp was executed with a minimum quality score of 20 and automatic adapter sequence detection (‘-q 20 –detect_adapter_for_pe’).

Reviewer comment:

The date or version of the NR database does matter.

Authors’ response: 

The access date of our local NR database, March 21st 2021, has been specified in the ‘Alignment’ section of Material and Methods.

Reviewer comment:

How the dictionary was made? Is the script or dictionary itself are available? Figshare or Zenodo could help.

Authors’ response:

The dictionary of GO terms was built using the createDictionary script provided with the MEDUSA pipeline (https://github.com/dalmolingroup/MEDUSA/blob/main/createDictionary.R). This dictionary was created using the ID mapping database provided by UniProtKB (ftp.uniprot.org/pub/databases/uniprot/current_release/knowledgebase/idmapping/), which is the source for their ID mapping service (https://www.uniprot.org/id-mapping). By using this dictionary and the annotate tool provided as a step within the MEDUSA pipeline, we could map the RefSeq identifiers obtained for the best alignment hits to Uniprot identifiers, which were then mapped to UniProtKB/Swiss-Prot Gene Ontology terms. These steps have been further clarified in the ‘Functional annotation’ section of  our methodology.

Reviewer comment:

L178, L212: how the clustering was performed in details? The raw data (distance matrices better to be published as Supplementary table).

Authors’ response:

We improved the Material and Methods section by describing the clustering analysis. We also added the dissimilarity matrix as Supplements (Supplementary table S2) 

Reviewer comment:

How can authors interpret very high percentage of viruses (Fig. 3B)?

Authors’ response: 

The high percentage of viruses is mostly associated with groups G1-G3, which comprehend samples in the two most superficial layers (SRF and DCM), mostly within the Indian ocean. This finding could be associated with earlier studies, such as Lara et al, 2017, which showed that viral abundance decreased 10-fold with depth in tropical and subtropical regions - which were the focus of our analysis. Additionally, the same study affirmed that the highest level of viral abundance was detected in the mesopelagic layer of Indian Central Water, which is where most samples from G1-G3 were also extracted. We have better detailed this possible association in our discussion.

Reviewer comment:

The signs of domains (Archae, Bacteria, Viruses) are hard to interpret and intersecting with the tree. Sample ID’s are missing on the Figure 3.

Authors’ response:

We altered the colors for each domain in the Figure 3A to avoid them to be mistaken for the layer colors  on the right-side boxplots. Also in Figure 3A, we changed the sign representing the oceans to colored rectangles to avoid them to be mistaken for the signs for each layer. We have added the sample names on the bar plot. 

Reviewer comment:

Colors of the same terms at different panels are different, which makes them harder to compare.

Authors’ response:

Following the other Reviewer’s suggestions, we changed the visualization of the GO terms to a heatmap, which makes the comparison of term frequency in different layers easier.

Reviewer comment:

L285: the reference [35] doesn’t contain the determination of “oceanic area below 2000 m deep”, please cite the original source.

Authors’ response:

We added the correct reference for the sentence. 

Reviewer comment:

L303: this manuscript describe the microbiome of “tropical or subtropical regions”, comparison with Arctic region looks strange here.

Authors’ response:

We agree with the reviewer that comparisons between tropical and subtropical regions with arctic and antarctic ones can be misleading or confusing. However, since in this paragraph we are comparing the vertical variation patterns in abundance and diversity of microorganisms of our data to the findings of these earlier studies, we still think this comparison is valid. Jing et al, 2013 found that depth variations had a similar impact on diversity in three distinct latitudes of the Pacific Ocean, which included subarctic and tropical regions. Given these findings and the limited amount of studies on the influence of depth in microorganism composition in tropical and subtropical regions, we chose to maintain the comparison. However, we have addressed the reviewer’s concerns by adding a section explaining the reason for this comparison.

Reviewer comment:

L316: virus taxonomy data are missing.

Authors’ response: 

Following this and the other Reviewer’s suggestion, we have included Supplementary Figure 2 as part of Figure 3 (Figure 3C), which now shows the taxonomic composition of the samples. By presenting in this figure that the most abundant viruses come from uncultured taxa, we support the findings described by Liang et al, 2019, discussed on line 316.

Reviewer comment:

L319: the reference [51] describes the viral particles extraction protocol only, not the cause of “diversity of aquatic viruses remains largely unexplored”.

Authors’ response:

We thank the Reviewer for noticing this. We added three references that summarise the current state-of-the-art knowledge of Virus identification in aquatic environments. 

Reviewer comment:

Some minor corrections to the text (style and spelling):

  • L271, 273: repetition, please rephrase
  • The references (24, 26, 47, 50) have missing journal name/DOI.

Authors’ response:

We thank the Reviewer for noticing these mistakes. We rephrased the sentence from lines 271-273 to improve clarity. Also, we added the DOI code for the mentioned references, except for reference 26, which stands for the Fastx-Collapser tool and does not have an associated publication. 

Round 2

Reviewer 1 Report

Thank you for making the required changes.

Author Response

We again thank the valuable suggestions and comments provided by the Reviewer.

Reviewer 2 Report

No additional comments

Author Response

We would like to thank the Reviewer again for the valuable comments and suggestions.

Reviewer 3 Report

Thank you very much for the comprehensive answers!

However, some of the questions still bother me.

1. In Figure 3A, the sample names are not associated with the station names.

2. Nevertheless, it remains unclear why several replicates were made for some deep horizons, and for some only one. Does it probably affect for further statistics?

Manuscript has not proven that only depth affect the biodiversity. It seems to me that this is a set of factors. If this is not the case, it requires proof (probably an analysis of the relationship of biodiversity with physicochemical parameters).

Author Response

REVIEWER 3
Reviewer comment:
Thank you very much for the comprehensive answers!
However, some of the questions still bother me.
1. In Figure 3A, the sample names are not associated with the station names.
Authors’ response:
We would like to thank the Reviewer again for the great comments and suggestions which
helped to improve the paper. Actually, Figure 3A already has the station names. The sample
names in Figure 3A are a combination of the station and the accession number. (e.g., the
T064 - 594324 sample belongs to the station 064, and the ENA accession number is
ERR594324). To make it as clear as possible, we added an observation concerning the
station names in the figure captions. We also added another column on the Supplementary
Tables 1 and 4 explaining the sample names.
Reviewer comment:
2. Nevertheless, it remains unclear why several replicates were made for some deep horizons,
and for some only one. Does it probably affect for further statistics?
Authors’ response:
As we state in the text, we selected the stations in the tropical and subtropical regions that
have at least one sample by depth layer. We used all data available according to our criteria.
As the minimal number of samples in a station and in a depth layer is one, downsampling
other stations and layers would impact the test’s statistics, but in a negative way because we
would be losing important information. Regarding the specific reasons why researchers
collected a different number of samples among stations/layers, the method’s paper (Pesant et
al., https://doi.org/10.1038/sdata.2015.23) is not clear.
Reviewer comment:
Manuscript has not proven that only depth affect the biodiversity. It seems to me that this is a
set of factors. If this is not the case, it requires proof (probably an analysis of the relationship
of biodiversity with physicochemical parameters).
Authors’ response:
We agree with the Reviewer comment and thank him/her for the insightful consideration. Our
intention was not to prove a causal relationship between the depth and the biodiversity. We
only identified that there are significant diversity differences when comparing the three
layers. We added an observation about that in the first paragraph of the Discussion section.

Reviewer 4 Report

I would thank authors for thorough revision and answers to my comments. The obtained results are interesting, but hard to reproduce. I recommend to prepare detailed instructions for reproducing authors` results. The repository "meta_taraoceans" requires readme file with description of its contents. 

The detailed manual will help to use (an cite!) your tool (MEDUSA). 

Author Response

REVIEWER 4
Reviewer comment:
I would thank authors for thorough revision and answers to my comments. The obtained
results are interesting, but hard to reproduce. I recommend to prepare detailed instructions for
reproducing authors` results. The repository "meta_taraoceans" requires readme file with
description of its contents.
The detailed manual will help to use (an cite!) your tool (MEDUSA).
Authors’ response:
We again would like to thank the Reviewer for providing insightful considerations. We
completely agree with the Reviewer regarding the GitHub repository and we have provided a
detailed description for each directory and files on the meta_taraoceans repository. In the
main README file, we reference the step-by-step guide of the MEDUSA pipeline, which was
used in our analysis and can be found in the Supplementary Material in the MEDUSA paper
(https://www.frontiersin.org/articles/10.3389/fgene.2022.814437/full#supplementary-material
)
